# Great Potential of Si-Te Ovonic Threshold Selector in Electrical Performance and Scalability

**DOI:** 10.3390/nano13061114

**Published:** 2023-03-21

**Authors:** Renjie Wu, Yuting Sun, Shuhao Zhang, Zihao Zhao, Zhitang Song

**Affiliations:** 1State Key Laboratory of Functional Materials for Informatics, Shanghai Institute of Micro-System and Information Technology, Chinese Academy of Sciences, Shanghai 200050, China; 2University of Chinese Academy of Sciences, Beijing 100029, China

**Keywords:** OTS material, high scalability, high on-current density, PF model

## Abstract

The selector is an indispensable section of the phase change memory (PCM) chip, where it not only suppresses the crosstalk, but also provides high on-current to melt the incorporated phase change material. In fact, the ovonic threshold switching (OTS) selector is utilized in 3D stacking PCM chips by virtue of its high scalability and driving capability. In this paper, the influence of Si concentration on the electrical properties of Si-Te OTS materials is studied; the threshold voltage and leakage current remain basically unchanged with the decrease in electrode diameter. Meanwhile, the on-current density (*J*_on_) increases significantly as the device is scaling down, and 25 MA/cm^2^ on-current density is achieved in the 60-nm SiTe device. In addition, we also determine the state of the Si-Te OTS layer and preliminarily obtain the approximate band structure, from which we infer that the conduction mechanism conforms to the Poole-Frenkel (PF) model.

## 1. Introduction

The fact that the digital data are exponentially increasing at every moment has put forward a series of higher requirements for memory—more reliability with less cost, greater memory capacity with smaller sizes, and faster read/write speed with lower power consumption [1,2]. Compared to NAND Flash devices, PCM has been one of the most promising competitors of the next-generation memories on the strength of structural transition of chalcogenides between crystalline and amorphous states [3,4] which has shown extraordinary advantages in operating speed, storage density and endurance [5]. Moreover, the successful commercialization of PCM [6] is also inseparable from the help of ovonic threshold switching (OTS), which tends to suppress leakage current in case of operating errors. Compared with traditional Si-based switches, OTS materials acting as just nm-scale films dispense with the share of Si substrate and possess highly potential in scalability [7].

However, loads of materials with superior performance emerging from the OTS research in full swing contain eco-unfriendly elements such as sulfur [8,9,10], arsenic [11,12] etc.; additionally, most OTS materials are ternary, quaternary or even more [13,14], which means they inevitably suffer from element segregation and inhomogeneous composition. At the same time, OTS switches are required to maintain an amorphous state at the high temperature of 400–450 °C, which is also called back-end-of-line (BEOL) process [1,15], posing a serious challenge to binary Se- and Te-based materials [16,17]. In 2021, a major breakthrough was made by Shen et al., in that tellurium was discovered to be the simplest threshold switching material with an endurance of 10^8^ cycles. The selectivity is higher than 10^3^ and the on current density is more than 11 MA/cm^2^, which is enough to drive the incorporated PCM. Different from previous OTS materials, the Te layer stays in a crystalline state before and after pulse operation; therefore, Shen et al. creatively raised a brand-new principle called the ‘Crystalline-Liquid-Crystalline Mechanism’, which corresponds the off state of the switch to the crystalline state of Te and then Te melts into liquid under the external voltage, which means the switch turns on. At the removal of applied voltage, Te rapidly recrystallizes and the device turns off. In addition, the single element also avoids the limitations from the high temperature of the BEOL process and the element/phase segregation [1].

Unluckily, the leakage current at 1/2 *V*_th_, i.e., *I*_off_, of 60 nm Te device is only 0.5 μA according to Shen’s research, which indicates the subthreshold region of Te is composed of two parts; one is dominated by the Schottky barrier between Te and TiN electrodes in low electric field, and the other depends on the intrinsic excitation of Te in higher field. Therefore, the ways to optimize the *I*_off_ lie in two perspectives: one is to increase the Schottky barrier by changing the electrode material, such as Pt; the other is to increase the band gap of the material including Si [18,19,20] or B [21] doping.

In our work, we carry out electrical tests on SiTe-based devices to explore the influence of different Si contents on device performance and their potential to scale down. Then we study the conduction mechanism of SiTe devices through characterization.

## 2. Materials and Methods

### 2.1. Film Preparation and Testing

Four compositions are designed to be SiTe, SiTe_2_, SiTe_4_ and SiTe_8_. All the films were deposited by co-sputtering of Si and Te targets, while the power values of Si target are 65 W, 40 W, 30 W and 20 W with 10 W of Te target. The exact components were determined by energy-dispersive spectroscopy (EDS). 100 nm-thick films were deposited on SiO_2_ substrates for in situ resistance-temperature (R-T) tests with 20 °C/min heating rate and X-ray diffraction (XRD) tests ranging from 10–67°. The Fermi level of different films were measured by X-ray photoelectron spectroscopy (XPS), and 400-nm Si-Te films were deposited on quartz glass as the ultraviolet-visible Spectrophotometer (UV-Vis) samples.

### 2.2. Device Fabrication

The 10 nm Si-Te OTS layers were deposited on TiN bottom electrodes whose diameters are 200, 150, 120 and 60 nm, respectively. Then, TiN was sputtered as the top electrode with 40-nm thickness. Keithley 4200A-SCS parameter analyzer was used to measure the electrical performance. Tektronix MSO54 mixed signal oscilloscope was used to capture the input pulses and device responses.

## 3. Results

### 3.1. Variations of Electrical Performance with Different Si Concentrations

T-shaped device structure is shown in Figure 1a where cylindrical TiN serves as the bottom electrode and Si_3_N_4_ acts as the isolation. Then current-voltage (*I-V*) curves of 200 nm-electrode devices with all four Si-Te compositions under 10 continuous triangular pulses (Figure 1b) are shown in Figure 1c. The red lines represent the response of device under the first pulse, which is called first fire (FF). During the process, SiTe device suddenly turns to low resistance state at 2.83 V, while under the second pulse, the threshold voltage (*V*_th_) goes down to 1.71 V as described by the blue lines. With the decrease of Si content, the fire voltage (*V*_fire_) of SiTe_2_ drops to 2.09 V. After FF process, only 1.11–1.31 V *V*_th_ is needed to operate the cell. The downward trend continues with SiTe_4_ and SiTe_8_, whose *V*_fire_ were 1.79 V and 1.63 V, respectively. Moreover, *V*_th_ of SiTe_4_ is ~0.8 V lower than its *V*_fire_ which fluctuates between 0.99 and 1.13 V, while the *V*_th_ of SiTe_8_ ranges from 0.89 V to 1.05 V. The amplitude of the first pulse applied to the SiTe device is 4 V and the height of the remaining nine pulses is 3 V. Meanwhile, the pulse amplitude of all ten triangular pulses used for other Si-Te devices is 4 V. The rising edges, falling edges and pulse intervals of these pulses are 1 μs. Clearly, both *V*_fire_ and *V*_th_ keep decreasing with the continuous decrease of Si contents.

Afterwards, we carried out DC *I-V* tests and the circuit is shown in Figure 2a. From Figure 2b, we can obviously find that the change trend of *V*_th_ is consistent with the results of pulsed *I-V*, that is, the addition of silicon will lead to the rise of *V*_th_. As can be seen in the figure, all Si-Te devices show >10 mV/dec nonlinear characteristics and more than 10^3^ selective ratio. In addition, the leakage current of SiTe at ~1/2 *V*_th_, namely *I*_off_, which is about 0.12 μA, and the *I*_off_ of SiTe_2_ is situated at ~93 nA. It continued to decrease to 53 nA by the concentration of Si declines to 20 at. %, but contrary to the down trend of Si concentration, *I*_off_ increases to 0.15 μA in SiTe_8_. However, after summarizing and comparing the DC curves of the four components as shown in Figure 2c, we clearly observed that - the leakage current decreases with the increase of Si content.

Other electrical performances are shown in Figure 3. Instantaneous responses of the device are observed through the oscilloscope, from which we are able to capture the moment when the device is turned on and off, as shown in Figure 3a. The abrupt rise of current signifies that the device is switched on, while the time span is considered to be the on-speed. Similarly, a sudden drop in current refers to the off-speed. As shown in Figure 3b, the content of silicon seems independent of the switching speeds. Moreover, the lifetime of the devices was measured with the help of the oscilloscope. A fixed number of continuous pulses is applied to the device, and whether the device still works normally is judged from the screen. In case the device responds to each pulse, a DC test is conducted next to measure its *I*_off_ in order to verify whether the off-state is in a high-resistance state as before. Only if both conditions are met, the endurance with that fixed number is proved. Then the number of input pulses gradually increases until the device keeps silent to the input or the *I*_off_ of the device goes too large. In this way, we find the endurance of both SiTe and SiTe_8_ are merely 10^5^, but the reason for that low value is slightly different. SiTe is unable to be switched on after 10^5^ pulses. As for SiTe_8_, it could be turned on but its *I*_off_ is too large, which may be due to the low crystallization temperature. As shown in Figure 3c, SiTe2 device enables to operate for 108 cycles which shows the longest lifetime among these four contents.

### 3.2. Effect of Device Size on Electrical Performance

From the perspective of market demand, device shrinkage has been focused, and whether the device maintains the performance with scaling down has been a major problem. First, we explored the relationships between *V*_fire_, *V*_th_, holding voltage (*V*_h_) and electrode sizes. In fact, *V*_fire_ significantly affects the device lifetime and power consumption, while *V*_th_ and *V*_h_ are directly related to the read margin [22], that is, a slight shift of *V*_th_ or *V*_h_ may cause serious operation errors. Luckily, Si-Te series devices show high potential in scalability from Figure 4a. According to the statistics of more than 50 independent cells, *V*_fire_, *V*_th_ and *V*_h_ are almost irrelevant to the device size. Clearly shown in Figure 4a and Figure 1c, the read margin becomes narrow with the decrease of Si content, but the good news is that *V*_fire_ also becomes smaller. Therefore, the key to application is how to balance these two points. In addition, the reduction of device size brings the increase of on-current density as well. Generally, >10 MA/cm^2^ is required to drive the PCM. From Figure 4b, the on-current density of 60 nm-SiTe device reaches 25 MA/cm^2^ at most, and *J*_on_ of the other components also exceed 10 MA/cm^2^. As for *I*_off_, it hardly changes with the electrode size according to Figure 4c. The discrepancy of SiTe_8_ results from the number of devices participating in statistics. Under comprehensive consideration, SiTe_2_ is equipped with >15 mA/cm^2^ on-current density, 10^8^ cycles of endurance, and moderate *I*_off_ among these compositions, which is the best choice of the four components.

### 3.3. Characterization of Si-Te Films

We first carried out XRD tests on the as-deposited films of each component, as shown in Figure 5a. As seen from the figure, Si-Te films of the four components show no peak, but there are obvious crystallization peaks in the Te sample, indicating that the deposited Te is in crystalline state, while the state of film turns to amorphous after Si incorporation. At the same time, the state of the material has also been proved by R-T test in Figure 5b. Interestingly, the crystallization temperature of SiTe is only 78 °C, and the crystallization temperature reaches the peak of ~235 °C at SiTe_2_. However, with the further reduction of Si content, the crystallization temperature decreases, and that of SiTe_8_ decreases to 128 °C, the tendency which is very similar to Ge-Te system [23]. Moreover, the different states directly indicate that the conduction mechanism of Si-Te is different from that of Te, while the Si-Te device is more consistent with the traditional PF model [24], where the switching characteristic strongly depends on the amorphous state. Then the location of the valence band maximums (VBM, E_v_) are determined by linear extrapolation of the linear part of the XPS valence band spectrum, they are situated at −0.48 eV, −0.45 eV, −0.41 eV and −0.39 eV for SiTe, SiTe_2_, SiTe_4_ and SiTe_8_, respectively, in Figure 5c. Meanwhile, the band gap is obtained by linear fitting according to αhν = C (hν − E_g_)^2^, where C is the constant, and α is the absorption coefficient, as shown in Figure 5d. The band gap value of SiTe is 1.02 eV. With the decrease of Si concentration, the band gap slightly increases to 1.07 eV, and further drops to 0.88 eV and 0.87 eV for SiTe_4_ and SiTe_8_. The Fermi level is basically located in the center of the band gap, which is generally believed that the high concentration defect states would pin the Fermi level in the middle of the band gap. Si-Te series materials basically conform to this rule, that is to say, their conduction mechanism may be closely related to the defect state, which requires further research.

## 4. Conclusions

In this paper, we comprehensively studied the relationships between the electrical properties of Si-Te devices and Si concentration. Although the operation speed of the Si-Te device is maintained at ~10 ns, *V*_fire_ and *V*_th_ gradually rise and *I*_off_ drops as the Si concentration goes up. However, the steady increase of Si concentration also leads to the lifetime degradation when the Si concentration exceeds 33 at. %, where the SiTe_2_ device exhibits a superior endurance of 10^8^ cycles to other contents. In addition, *V*_th_, *I*_on_ and *I*_off_ of all components hardly change with the electrode size, and the *J*_on_ of 60-nm SiTe device even increases to 25 MA/cm^2^, indicating the great potential of Si-Te devices in scaling down. In terms of the conduction mechanism of Si-Te, we firstly find that Si-Te materials are amorphous by XRD, and observe the change in trend whereby the crystallization temperature increases and then decreases with the decrease of Si contents through in situ RT experiment. Through UV-vis, we also find that the root of *I*_off_ variation with Si concentration originates from the band gap, that is, the larger the content of Si, the wider the band gap, and the smaller the *I*_off_. Afterwards, the Fermi level is found to be pinned at the center of the band gap through XPS, indicating that the conduction mechanism is more consistent with the traditional PF model associated with trap state which is different from Te switching. The specific relationship between the switching performance of Si-Te materials and the defect state is also worthy of further study.

## Figures and Tables

**Figure 1 nanomaterials-13-01114-f001:**
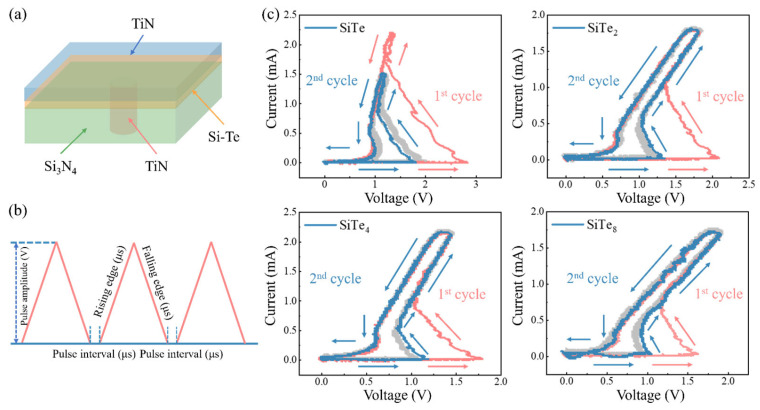
Pulse *I-V* curves of devices with different Si contents. (**a**) The schematic diagram of device structure. (**b**) Triangular pulses applied to the devices with modifiable rising edge time, falling edge time, pulse amplitude and interval. (**c**) *I-V* curves of devices with different compositions under 10 consecutive triangular pulses. Dotted lines correspond to FF, and the dashed lines refer to the second operation.

**Figure 2 nanomaterials-13-01114-f002:**
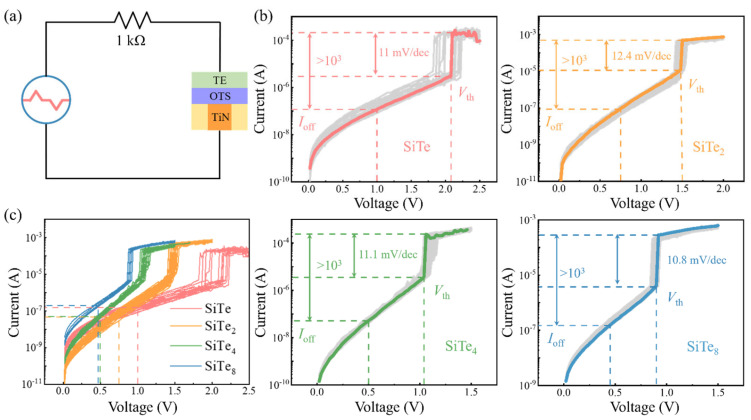
DC curves of devices with different Si contents. (**a**) A 1 kΩ resistor is in series with the device in the DC testing circuit. (**b**) DC *I-V* curves of Si−Te devices. (**c**) The synthesized DC curves of four compositions.

**Figure 3 nanomaterials-13-01114-f003:**
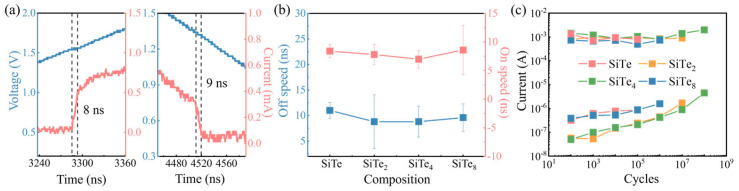
Speed and endurance distribution of Si–Te devices. (**a**) The on–speed and off−speed are 8 ns and 9 ns, respectively. (**b**) Statistical operating speeds distribution of different Si–Te devices. (**c**) Device lifetime of devices with all four compositions.

**Figure 4 nanomaterials-13-01114-f004:**
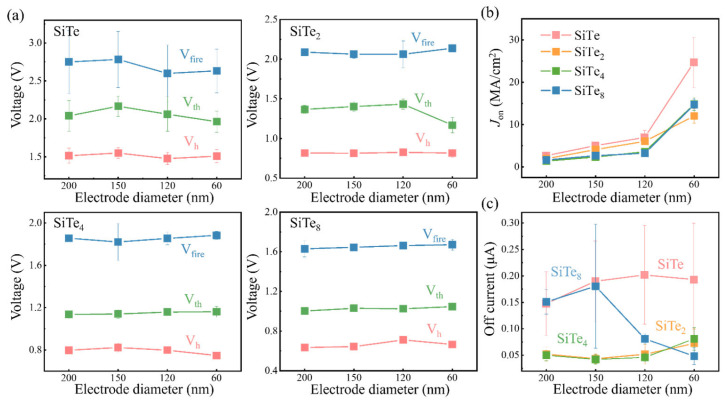
Device performances along with the device size. (**a**) *V*_fire_, *V*_th_ and *V*_h_ variations of SiTe, SiTe_2_, SiTe_4_ and SiTe_8_ with different electrode diameters. (**b**) Change trend of average on-current density with electrode sizes. (**c**) Tendency of average *I*_off_ with different electrodes.

**Figure 5 nanomaterials-13-01114-f005:**
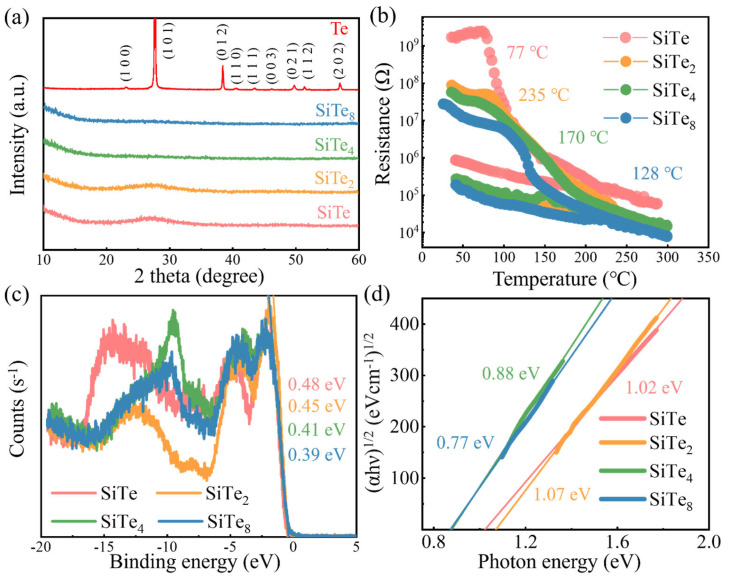
Characterization of Si−Te films. (**a**) XRD results of SiTe, SiTe_2_, SiTe_4_, SiTe_8_ and Te films. (**b**) R-T results of films with four Si−Te compositions. (**c**) XPS valence band spectrum of SiTe, SiTe_2_, SiTe_4_, SiTe_8._ (**d**) (αhν)^1/2^ versus hν interprets the bandgap of four compositions.

## Data Availability

Not applicable.

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
