# Peer review of "Great Potential of Si-Te Ovonic Threshold Selector in Electrical Performance and Scalability"

_nanomaterials, 2023, doi:10.3390/nano13061114_

Round 1

Reviewer 1 Report

1. hope the author can add pulse width (uS) and amplitude(mV) in figure 1a.

2.  Hope the author can polish figure 1c to better distinguish 1st cycle and 2nd cycle. dotted and solid line are very difficult to be seen. 

3. in Line143, it says that fermi level is located at the center of bad gap. how to get the centered fermi level from Ev and Eg?also,  the doubt is that in high defects density, should the fermi level be pushed to Ec or Ev side when it is n-type or p-type?

Author Response

We have revised our manuscript and answer your questions in attachment. I hope our responses satisfy all of your concerns. Please see the attachment.

Reviewer 2 Report

The authors study the influence of Si concentration on the electrical properties of Si-Te OTS materials. The objective is to test SiTe-based devices to explore the influence of different Si contents on device performance.

The paper is generally well written and structured and has solid merits for publication. As a matter of fact, I only have a few minor remarks to be addressed by the authors:

- In a few figures, the font size should be a bit larger for better readability, for instance Fig.  4c, Fig. 3, etc. 

- In Fig. 4, it is not quite clear what is part a, part b and part c of the figure. This should be better arranged. 

- Similarly in Fig. 5, letter c is missing (presumably the down left part of the figure). 

- Conclusions are quite sparse and basically only repeat what has been done in the paper. Conclusions should go further and provide some general conclusions originating from the work. Also, the ideas for the future work should be addressed more specifically. 

- Finally, it should be noted that the authors have not properly used the journal template and this should be corrected as well. 

Author Response

(The authors gave the same response as above.)
